# Deep learning based ECG segmentation for delineation of diverse arrhythmias

**Chankyu Joung[1]☯, Mijin Kim[2]☯, Taejin Paik[1], Seong-Ho Kong[3,4], Seung-Young Oh[4,5], Won Kyeong Jeon[6], Jae-hu Jeon[7], Joong-Sik Hong[7], Wan-Joong Kim[7], Woong Kook[1,3], Myung-Jin Cha[2]*, Otto van Koert[1]***

**1** Department of Mathematical Sciences and Research Institute of Mathematics, Seoul National University, Gwanak-gu, Seoul, South Korea, **2** Division of Cardiology, Department of Internal Medicine, Asan Medical Center, University of Ulsan College of Medicine, Seoul, South Korea, **3** AI Institute, Seoul National University, Seoul, South Korea, **4** Department of Surgery, Seoul National University Hospital and Seoul National University College of Medicine, Seoul, South Korea, **5** Department of Critical Care Medicine, Seoul National University Hospital, Seoul, South Korea, **6** Department of Internal Medicine, Seoul National University Hospital, Seoul, South Korea, **7** Medifarmsoft Co., Ltd., Seoul, South Korea

☯ These authors contributed equally to this work.
* chamj81@gmail.com (MJC); okoert@snu.ac.kr (OK)

**Data Availability Statement:** The internal database cannot be shared publicly because the IRB of Seoul National University Hospital hasn't cleared this data (anonymized ECGs) for publication. Data are available from the Seoul National University

## Abstract

Accurate delineation of key waveforms in an ECG is a critical step in extracting relevant features to support the diagnosis and treatment of heart conditions. Although deep learning based methods using segmentation models to locate P, QRS, and T waves have shown promising results, their ability to handle arrhythmias has not been studied in any detail. In this paper we investigate the effect of arrhythmias on delineation quality and develop strategies to improve performance in such cases. We introduce a U-Net-like segmentation model for ECG delineation with a particular focus on diverse arrhythmias. This is followed by a post-processing algorithm which removes noise and automatically determines the boundaries of P, QRS, and T waves. Our model has been trained on a diverse dataset and evaluated against the LUDB and QTDB datasets to show strong performance, with F1-scores exceeding 99% for QRS and T waves, and over 97% for P waves in the LUDB dataset. Furthermore, we assess various models across a wide array of arrhythmias and observe that models with a strong performance on standard benchmarks may still perform poorly on arrhythmias that are underrepresented in these benchmarks, such as tachycardias. We propose solutions to address this discrepancy.

## 1 Introduction

An electrocardiogram (ECG) is a basic medical diagnostic tool that monitors the electrical activity of the heart. It is non-invasive, relatively quick to perform, and inexpensive while providing a wealth of information about the overall health of the heart. Traditionally, the analysis of the structural elements in an ECG, including the durations and morphology of the QRS complex, the P and T waves (see Fig 1), plays a key role in identifying abnormalities or irregularities in the heart's electrical activity that may point towards underlying heart conditions [1].

Hospital Institutional Data Access / Ethics Committee The contact information of the IRB of SNUH is as follows: Seoul National University Medical College/Seoul National University Hospital Medical Research Ethics Review Committee Tel: 82-02-2072-0694/2266 FAX: 82-02-3675-6824 03080, 101 Daehak-ro, Jongno-gu, Seoul, Republic of Korea http://hrpp.snuh.org/irb/introirb/_/singlecont/view.do Email requests for data access can be sent toirb@snuh.org or to Myung-Jin Cha, chamj81@gmail.com The other data underlying the results presented in the study are available from Physionet, specifically QTDB, LUDB, MIT-BIH and PTB-XL. These public databases are available at https://physionet.org/content/qtdb/1.0.0/ https://physionet.org/content/ludb/1.0.1/ https://physionet.org/content/mitdb/1.0.0/ https://physionet.org/content/ptb-xl/1.0.3/.

**Funding:** This work was supported by the Korea Medical Device Development Fund grant funded by the Korea government (the Ministry of Science and ICT, the Ministry of Trade, Industry and Energy, the Ministry of Health \& Welfare, the Ministry of Food and Drug Safety) (Project Number: 1711174270, RS-2021-KD000008), (JSH, JHJ). https://www.msit.go.kr/eng/index.do In addition, WK and OvK receive support from the National Research Foundation of Korea (NRF) Grant 2022R1A5A6000840 (joint), as well as (MSIP) [RS-2022-00165404 (WK), NRF2023005562 (Ovk)], funded by the Korean Government. https://ernd.nrf.re.kr/index.do None of the funders played a role in the study design, data collection and analysis, decision to publish, or preparation of the manuscript.

**Competing interests:** I have read the journal's policy and the authors of this manuscript have the following competing interests: CJ, TP, WK and OvK received financial support through NRF grants 2022R1A5A6000840 as well as RS-2022-00165404, NRF2023005562, funded by the Korean Government. SHK, SYO, JHJ, JSH, WJK and MJC received financial support through grants 1711174270, RS-2021-KD000008 funded by the Korean government. In addition, SHK, SYO, JHJ, JSH, WJK and MJC are Stockholders of Medifarmsoft Co., Ltd.

Precise delineation, which involves identifying the onset and offset of these waves and not just the peaks, is hence critical.

As such, automatic delineation of ECGs has been an important and well-developed topic, starting with rule-based techniques for locating the QRS complex, to wavelet transform-based delineation, [2–4] and deep learning techniques. Wavelet transforms deliver state-of-the-art performance in the benchmark QT database (QTDB), [5]. However, as [6, 7] point out, rule-based approaches typically require the adjustment of a threshold value for high scores, which may limit their generalizability to other datasets. Deep learning offers an alternative as shown for example in [6–8]: Jimenez-Perez et al. [6] used a U-Net type architecture [9] to achieve delineation performance comparable to wavelet-based methods on QTDB, and Moskalenko et al. [8] reported higher delineation performance compared to wavelet-based algorithms on the Lobachevsky University Database (LUDB) [10].

Although deep learning based delineation has shown excellent performance on benchmarks, arrhythmias pose a particular challenge in two important ways. First of all, there is a lack of data, reflected in the fact that the benchmark datasets are quite small and their samples tend to have relatively low heart rates. As a result, the variety of arrhythmias for the purpose of testing delineation quality is limited despite the careful preparation of these databases. A second important obstacle is that many arrhythmias cause significant changes in the structural elements and morphological features of an ECG. These changes are particularly striking in case of the P wave, which usually has the lowest signal to noise ratio. For example, in atrial fibrillation (AFIB) and atrial flutter (AFL) the P wave is absent, and a fibrillatory signal or flutter wave is found instead. As noted in [11, 12], false P wave predictions during such events present a significant challenge for delineation algorithms in clinical practice. Other arrhythmias, such as atrioventricular (AV) block, affect not only the position of P waves in relation to the QRS complex, but also their occurrence. This can result in P waves and QRS complexes following independent rhythms. In all of these cases, the performance of a P wave delineation algorithm is affected adversely. For instance, Aziz et al. [13] report a considerable drop in sensitivity for P wave detection in the case of ECGs exhibiting arrhythmia.

In this paper, we investigate this two-fold problem in the setting of deep learning and develop remedies. Specifically, we evaluate the performance of deep learning models, and find that the performance drops markedly in the presence of certain arrhythmia, such as various forms of tachycardia. An example of such failure is shown in Fig 6. Because tachycardia are underrepresented in both QTDB and LUDB, this drop doesn't affect the average performance of delineation in these benchmarks much, which explains why previous approaches still performed well on the benchmark tests. As remedies, we build on prior studies and devise a segmentation model with a U-Net like architecture to delineate ECG signals with diverse arrhythmias by training on a new dataset consisting of a large number of recordings with various arrhythmia types. In addition, we develop a segmentation model using a hybrid loss function that combines segmentation with the task of arrhythmia classification. This classification guided approach can ameliorate false P wave predictions for AFIB and AFL in short signals. A flow diagram of the model is shown in Fig 2.

The key contributions of this paper can be summarized as follows:

- Training a segmentation model that accurately delineates a chosen set of common arrhythmia types, achieved by using a diverse training set and employing a suitable post-processing strategy.

- Identifying common failure cases of segmentation models through separate validation on different arrhythmia types.

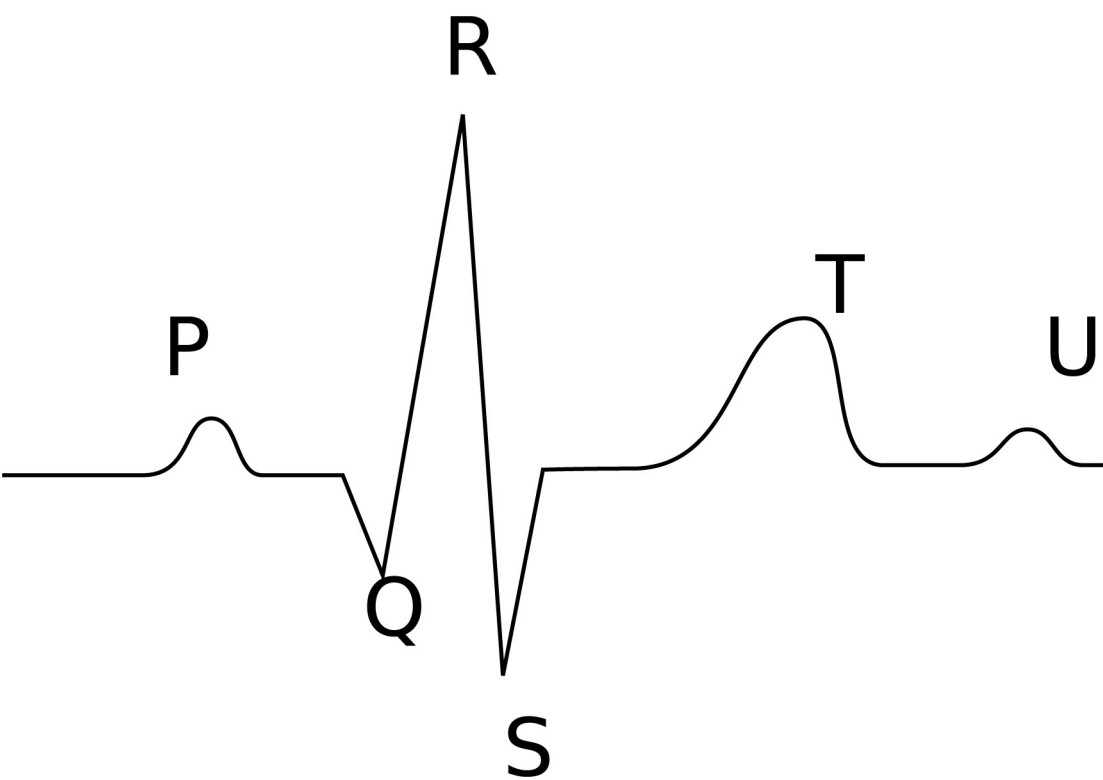

**Fig 1. A schematic representation of an ECG signal measured in lead I or lead II with the main complexes indicated.**

- Evaluating our model's performance on benchmark datasets QTDB and LUDB, demonstrating generalizability by results comparable with previous research.

- Introducing a classification guided strategy to reduce false P wave predictions for AFIB and AFL in short signals.

The rest of the paper is structured as follows. The Related Work Section 2 provides a review of relevant literature on ECG delineation and deep learning-based ECG analysis. The Methods Section 3 outlines the databases used for this study and presents the proposed delineation algorithm. The Results Section 4 presents performance evaluation metrics and reports experimental results. The Discussion Section 5 offers interpretations, implications, and discusses limitations and future directions. Finally, the Conclusion Section 6 concludes the paper.

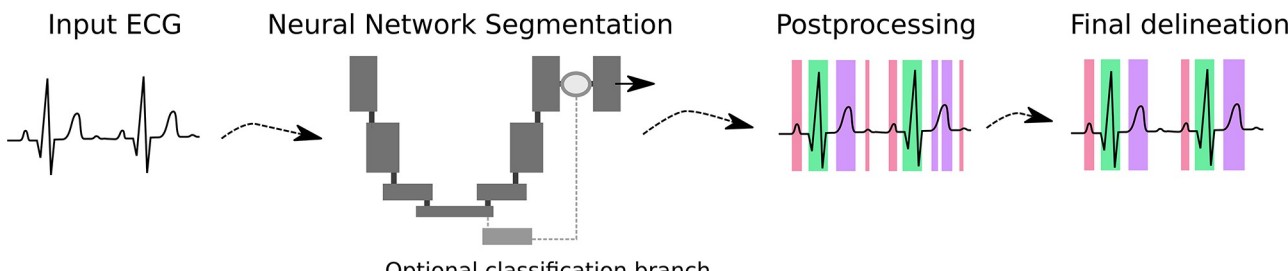

**Fig 2. Flow diagram for ECG delineation: An ECG input signal is segmented by a U-Net like model using an optional classification branch, and post-processed for noise, before producing final delineation results.**

## 2 Related work

### 2.1 Traditional approaches for ECG delineation

Early works on ECG delineation were primarily focused on developing rule-based methods to identify and locate the QRS complex. Pan and Tompkins [14] presented a seminal example of detecting the QRS complex by utilizing slope, amplitude, and width information. Subsequently, more advanced techniques have been employed to identify also the P and T waves. These include digital signal processing such as the wavelet transform [2–4, 15], the Hilbert transform [16, 17], and the phasor transform [18]. Additionally, classical machine learning approaches like hidden Markov models [19, 20] and Gaussian mixture models [21] have also been employed. Among these, wavelet-based methods have been widely cited as being the state-of-the-art, based on their delineation performance on public datasets such as QTDB and LUDB [4]. However, despite their effectiveness, traditional methods typically require manual feature extraction or domain-specific knowledge, whereas wavelet-based algorithms demand careful threshold selection for consistent results on different datasets.

### 2.2 Deep learning based ECG delineation

In recent years, deep learning has shown remarkable success in ECG processing such as arrhythmia classification [22–25], which led to its increasing popularity in various downstream tasks [26]. Deep learning has been adopted in ECG delineation as well, where a segmentation model with a CNN architecture is typically trained to locate the P, QRS, and T waves, which is then used to carry out the delineation task. The use of CNN architectures in segmentation tasks offers the advantage of automatically learning hierarchical features from the ECG signals [27], enabling the model to effectively identify and localize the relevant waveforms. Jimenez-Perez et al. [28] presented an adaptation of the U-Net architecture [9] to 1-dimensional data, while Sereda et al. [29] deployed an 8-layer convolutional network and studied the effects of using an ensemble of networks as opposed to using a single network for the segmentation. Moskalenko et al. [8] developed a U-Net-like architecture that achieved state-of-the-art performance on LUDB in terms of F1-score, when compared to previous deep learning approaches [29] and wavelet-based methods [4]. In a similar study, Jimenez-Perez et al. [6] again adapted a U-Net for segmentation but with added emphasis on regularization techniques for training with limited data. Their model, when cross-validated on QTDB, demonstrated comparable performance to those using digital signal processing techniques such as wavelet transforms [3]. Recently, Chen et al. [7] developed a 1D-U-Net model for classifying the sample points of a single heart-beat into P, QRS, T, and none categories. Together with their proposed post-processing strategy, the delineation algorithm outperformed other algorithms in terms of sensitivity for both QTDB and LUDB. Additionally, advanced architectures for ECG delineation models were explored in Nurmaini et al. [30] using convolutional recurrent networks, and in Li et al. [31] through an enhanced U-Net model combined with the transformer's encoder module.

### 2.3 Classification guided segmentation

In developing a neural network for semantic segmentation, it is sometimes beneficial to add an extra classification task. This approach has been particularly effective in the field of medical image segmentation, where detection of false positives is common for images in which the object of interest is not present. Huang et al. [32] addressed this problem of over-segmentation by introducing a classification guided module (CGM) where the model is trained with the additional classification objective of deciding whether or not a given image contains an organ.

By filtering out the segmentation output using the classification output, the number of false positives is reduced. A similar approach was taken by Shuvo et al. [33], where a separate localizer branch was added together with an additional classifier branch.

In the ECG literature, classification and segmentation tasks have remained separate for the most part, while deep learning architectures have shown great success for both tasks [26]. In our current work, we experiment with combining the two tasks by training an ECG segmentation model together with an additional arrhythmia classification learning objective. Previous studies have demonstrated the effectiveness of convolutional neural networks for arrhythmia classification. For example, Hannun et al. [22] trained a 34-layer convolutional neural network for arrhythmia classification of single-lead ECG signals, showing performance comparable to that of cardiologists. Ribeiro et al. [34] later used a residual network architecture, an architecture first developed by He et al. [35] in the context of image classification, for the reliable diagnosis of 12-lead ECG signals. For a detailed review of deep learning applications in arrhythmia classification, we refer to the systematic reviews conducted by Xiao et al. [36] and Ansari et al. [37].

## 3 Methods

### 3.1 Data

For this study, we have used both internal and external datasets to develop and test our algorithm. The internal database was used for training the segmentation model and assessing delineation accuracy across diverse arrhythmias. The standard public datasets QTDB and LUDB were used for external validation of our algorithm. The characteristics of these datasets are summarized in Table 1 and elaborated upon in subsequent sections.

### 3.2 Internal dataset

We have assembled an internal database of ECG signals from 1,557 patients by searching the electrocardiography database (GE MUSE, GE Healthcare, Waukesha, WI) in a single center (Seoul National University Hospital, Seoul, South Korea). In the process of ECG extraction, all personal information was anonymized, so the consent form was waived. This study was then approved by the institutional review board of the participating center (H-1906–163-1044, continued as D-2010–096-1165). Data extraction commenced on May 31, 2022 and was completed on December 16, 2022. During data extraction, some of the authors (MJK, SHK, SYO, MJC) had access to information that could identify individual participants.

Our intent was to collect a dataset in order to conduct experiments to elucidate the segmentation performance for signals during arrhythmia. To do so, we identified 155 subjects with atrial fibrillation (AFIB) and 59 with atrial flutter(AFL). Among the rest, arrhythmia types were identified for 490 subjects as normal sinus rhythm (NSR), 84 as sinus tachycardia (ST), 115 as bundle branch block (BBB), 197 as first degree atrioventricular block (AVB1) and 29 as ventricular tachycardia (VT). The remaining 428 subjects had other heart conditions (not arrhythmia in the usual sense), such as premature atrial contraction (PAC) or premature ventricular contraction (PVC). A summary can be found in Table 2.

**Table 1. Descriptions of signals and their annotations for each of the databases.**

| Data Source | # Recordings | Duration | Frequency | Leads | Boundary Annotations |
|---|---|---|---|---|---|
| Internal Database | 1557 | 10 seconds | 500Hz, 250Hz | 2 (I, II) | P, QRS, T on/offsets |
| QTDB [5] | 105 | 15 minutes | 250Hz | 2 | P, QRS on/offsets, T offsets |
| LUDB [10] | 200 | 10 seconds | 500Hz | 12 | P, QRS, T on/offsets |

**Table 2. Distribution of arrhythmia of internal database.**

| Arrhythmia type | AFIB | AFL | AVB1 | BBB | NSR | ST | VT | Other | Total |
|---|---|---|---|---|---|---|---|---|---|
| # Recordings | 155 | 59 | 197 | 115 | 490 | 84 | 29 | 428 | 1557 |

Initial selection of ECGs from the electrocardiography database was based on the presence of common clinically significant cardiac arrhythmias. After reviewing ECG records, we excluded ECGs where disagreement on the review result was found, as well as ECGs that were too noisy to interpret reliably. The original arrhythmia automatic diagnosis from the database, the commercial interpretation product, the MUSE Cardiology Information System by GE, confirmed by an overreader, was then independently reviewed by two expert cardiologists. Only when both readings were in agreement was it applied to the analysis. After that, the onsets and offsets for P, QRS, and T waves were manually annotated for each lead independently by a cardiologist using a custom-made software tool. The annotation results were then confirmed by another cardiologist. As a quality control measure, we include in the S1 Appendix statistics on the difference between lead I and lead II annotations.

For each subject, the extracted data consisted of a recording with a duration of 10 seconds for leads I and II with a sampling frequency of either 250Hz or 500Hz. The dataset was partitioned into a training set and a test set. The training set comprised 1032 recordings and was organized to include approximately 70% of recordings for each identified arrhythmia class. The test set was composed of the remaining 525 recordings.

### 3.3 The QT Database (QTDB)

The QT database (QTDB) [5] is a publicly available database that has been used widely for developing and evaluating ECG delineation algorithms, due to its inclusion of manual annotations. The database collects recordings from multiple databases including the MIT-BIH arrhythmia database [38], the European ST-T Database [39], and other databases to represent various QRS and ST-T morphologies. In total, there are 105 two-lead signals sampled at 250Hz with each signal lasting for 15 minutes. Manual annotations by cardiologists are included for at least 30 beats per record, which amounts to more than 3600 beats. The annotations include the peaks and boundaries of waveforms, and in particular include the onset and offset of the P wave, the onset and offset of the QRS complex, and the offset of the T wave. These annotations will be used to measure the delineation quality of our algorithm and to compare with previous wavelet based methods [4, 7, 40].

### 3.4 Lobachevsky University Database (LUDB)

The Lobachevsky University Database (LUDB) is a more recently published database, also developed as an open-access tool for validating ECG delineation algorithms. Unlike QTDB, LUDB consists of short signals of 10 seconds from 200 unique subjects, with 12-lead recordings sampled at 500Hz included for each subject. Furthermore, LUDB contains a complete set of annotations for all onsets and offsets of P, QRS, and T waves, which is included for each single lead signal. In particular, the total number of annotated beats is considerably higher than that of QTDB, and this large number of annotated single-lead signals has led studies to take advantage by using LUDB as training data for their ECG segmentation models [8, 29]. In this paper, we use LUDB for two purposes. First, we use LUDB alongside QTDB to validate our delineation algorithm and compare with existing methods [4, 8, 29]. Second, we study the delineation performance on various arrhythmias when the segmentation model is trained on LUDB as opposed to the diverse training set sourced from our internal dataset.

## 3.5 Overview of delineation algorithm

The proposed algorithm consists of two stages. The first is a segmentation stage where a single lead input signal is passed through a deep learning based segmentation model. As a result, the signal is segmented into intervals that belong to one of four types: P wave, QRS complex, T wave, or none of these. The second stage consists of post-processing in which the final decision on the onset and offset for each of the waveforms is made. The details of each stage are given in the following sections.

## 3.6 Segmentation model

We have adapted the encoder-decoder structure of U-Net [9] to our model in a similar fashion as in the previous papers [6, 8, 29] to work in the context of ECG signals. Namely, the original convolutions are replaced with 1D convolutions to work with time series data. We have further modified the structure by incorporating full-scale skip connections, and adding a separate classification branch whose role will be discussed in the Arrhythmia Classification Guidance Section 3.8. The resulting high-level architecture of our model is shown in Fig 3.

The encoder takes a single-lead ECG signal sampled at 500Hz as input and encodes it into five feature maps at multiple scales through a series of 1D convolutional blocks and MaxPooling layers which downsample by a factor 2. The decoder uses convolutional blocks and linear interpolation layers to transform these features into an output consisting of four channels of the same resolution as the input. As in the U-Net variants [32, 41], we allow the decoder networks to learn from and aggregate features coming from multiple levels by adapting the full-scale skip connections of [32]. The final segmentation output is obtained by passing the output of the decoder through a convolutional layer with 4 filters and kernel size 1 and applying a

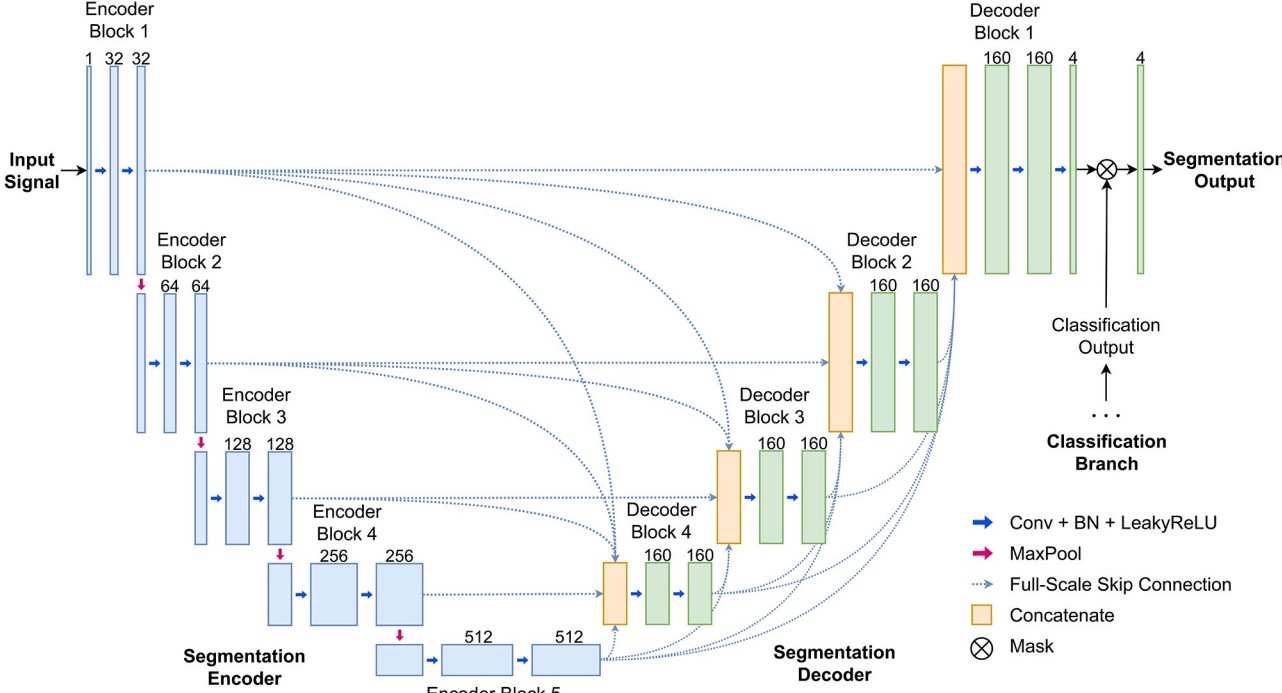

**Fig 3. Segmentation model architecture.** Our architecture is similar to U-Net3+, but uses 1D convolutional blocks and has an additional classifier branch.

softmax classifier for four classes: P wave, QRS complex, T wave, and none of these. This gives four class probabilities for each time stamp.

Note that for all other convolutional layers, we use a kernel size of 9 and padding of 4. As for the activation function, we use a leaky rectified linear unit with negative slope 0.01 for all layers. More specific details can be found in our implementation, which is available at https://github.com/ckjoung/ecg-segmentation.

## 3.7 Post-processing

The waveform boundaries are determined from the segmentation output through a post-processing stage, which consists of the following three steps. First, we extract segments of each type (P wave, QRS, T wave, none) by taking connected intervals where the probability of that type outputted by the model is highest. As a second noise reduction step, we discard short connected regions (of a duration less than 40 ms) and adjust the label based on the segmentation results of the adjacent intervals. In particular, we adjust the label according to the following rule:

1. if the two intervals adjacent to a short region have the same label, we regard the short segment as having the same label, thereby gluing the two regions to a single segment;

2. if the labels of the adjacent intervals are different, we discard the short region and label it as being none of the waveforms.

In the final step, we proceed by choosing the longest intervals labeled as P wave and T wave between consecutive QRS intervals and obtain their onsets and offsets. It can of course happen that there is no P wave, for example in the case of atrial fibrillation, or no T wave, which is very rare. This procedure automatically removes noise and returns unambiguous results.

## 3.8 Arrhythmia classification guidance

Here, we introduce an arrhythmia classification guided strategy for segmentation. The idea is to train the segmentation model jointly with a classification loss based on the arrhythmia type of each input signal. This is done by adding a classification branch following the deepest layer of the encoder, which predicts the arrhythmia type of the input signal. The weights of the model are affected by the joint training, and in addition, we can directly suppress the P wave segmentation output when the signal is predicted to belong to AFIB or AFL (see Fig 3). In the Reduced False P wave Predictions Section 4.4, we show that this approach can effectively reduce the number of false positive P wave predictions when delineating 10-second ECG signals. However, for other experiments, we only use the segmentation model without the classifier branch. Note that the proposed approach is similar to the classification guided modules of [32, 33], which have been used in the context of biomedical image segmentation. Here, we have re-designed the structure for the task of arrhythmia classification of ECG signals.

The structure of the arrhythmia classification branch is shown in Fig 4. The classification branch itself consists of two convolutional layers using 512 filters and a kernel size of 17. We

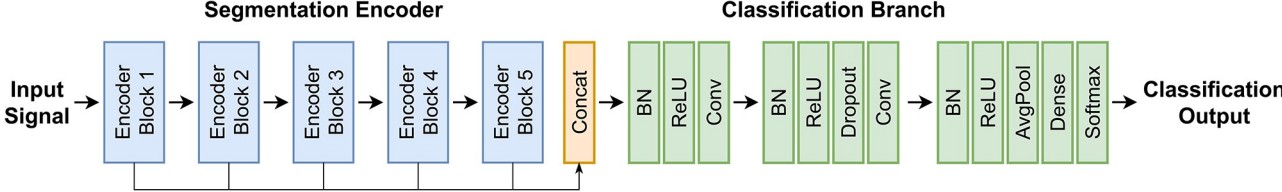

**Fig 4. Arrhythmia classification branch network architecture.**

apply batch normalization and dropout for regularization following the classification models of [22, 34]. The arrhythmia classification is performed by the final fully connected layer with softmax activation, whose output represents the probabilities of the signal belonging to either an AFIB or an AFL episode or not. A final prediction is made using an argmax function. Note that we have allowed the classification branch to take as input not just the feature of the last encoder block, but of encoder blocks of all levels. This is done by an aggregation scheme which works as follows. We first downsample the features of the first four encoder blocks to a size equal to that of the last encoder block. The downsampling is done using an average pooling layer. After the features have been resampled to the same shape, we concatenate the features to get a single aggregated feature.

### 3.9 Training

We have trained the network from scratch with convolutional weights initialized as in He et al. [42] using the Adam optimizer [43] with default parameters. The learning rate was initialized to be 0.001 and set to follow a cosine annealing schedule. To increase the diversity of training data, we applied data augmentation using transformations designed to mimic probable physiological noise, such as baseline wander and powerline noise, as used in [44]. The equations for these transformations are given as follows:

- Baseline wander:

$$n_{\mathrm{blw}}(t) = \sum_{k=1}^{50} a_k \cos(2\pi t k \Delta f + \phi_k) \tag{1}$$

- Powerline noise:

$$n_{\mathrm{pln}}(t) = \sum_{k=1}^{3} a_k \cos(2\pi t k f_n + \phi_1) \tag{2}$$

where $\Delta f = 0.01$Hz, $f_n = 50$Hz, with $a_k$ and $\phi_k$ uniformly sampled from $[0, 1)$ and $[0, 2\pi)$. We have also randomly resized the input signal by a factor $\exp(\alpha)$ where $\alpha$ is uniformly sampled from $[\log 0.5, \log 2]$, added random Gaussian noise with zero mean and standard deviation 0.01mV, and applied a constant baseline shift by an offset sampled from a Gaussian distribution. Fig 5 shows examples of the used transformations.

We adopt focal loss as introduced in [45] as our segmentation loss function. Focal loss modifies the standard cross-entropy loss by providing smaller weights to well-classified time stamps, letting the model focus on regions that are difficult to classify. The focal loss generalized to our multi-class segmentation setting can be written in the following form:

$$\mathcal{L}_{\mathrm{focal}} = -\frac{1}{N} \sum_{n=1}^{N} \sum_{c=1}^{C} (1 - \hat{y}_{n,c})^{\gamma} y_{n,c} \log \hat{y}_{n,c} \tag{3}$$

Here, $\hat{y}_{n,c}$ denotes the predicted probability of time stamp $n$ belonging to class $c$, while $y_n$ is the one-hot vector of the true class label for time stamp $n$. In our experiments, we use the default value of $\gamma = 1.0$. During arrhythmia classification guidance of the Arrhythmia Classification Guidance Section 3.8, we use the standard binary cross-entropy loss $\mathcal{L}_{\mathrm{bce}}$ for the classification branch. This gives the overall loss function:

$$\mathcal{L}_{\mathrm{total}} = \mathcal{L}_{\mathrm{focal}} + \alpha \mathcal{L}_{\mathrm{bce}}. \tag{4}$$

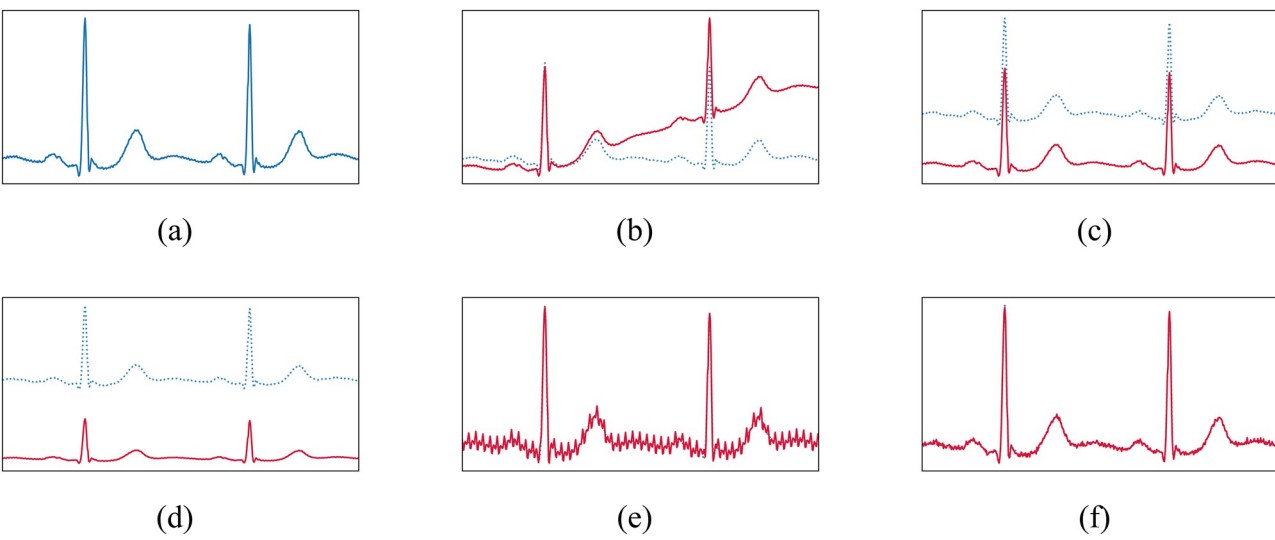

**Fig 5. Examples of transformations used for data augmentation.** (a) Original, (b) baseline wander, (c) baseline shift, (d) resize, (e) powerline noise and (f) Gaussian noise.

The additional trade-off parameter $\alpha$ can be adjusted to balance the effect of classification and segmentation losses during training. For all our experiments, we used $\alpha = 1$.

We train and validate our model using single lead ECG signals. To prevent potential issues arising from incomplete annotations for waveforms near the beginning and the end of a signal, we proceed as in [8] to exclude the initial and final 2 seconds of our signals during the training process. Hence, our model performs segmentation and classification using a signal of duration 6 seconds during training, and of 10 seconds during validation. While this scheme was designed mainly due to its practicality, we note that ECG recordings of 5 or 10 seconds have been shown to be successful for a CNN based arrhythmia classification [46]. We only use signals from leads I and II for training and validation of our model. Each input signal is resampled to 500Hz.

## 4 Results

### 4.1 Evaluation metrics

In order to evaluate the performance of the proposed delineation algorithm, we compare the ground truth annotations for the onsets and offsets of P, QRS, and T waves with the predicted annotations. We follow the usual standard chosen by The Association for the Advancement of Medical Instrumentation(AAMI) [47], which considers an onset or an offset to be correctly detected if an algorithm locates the same type of annotation in a neighborhood of 150ms. Using this threshold value, we examine for each predicted point whether the prediction correctly detects a point in the ground truth annotation.

If a ground truth annotation is correctly detected, we count a true positive(TP). In this case, the error is measured as the time deviation of the predicted point from the manual annotation. If there is no point of the ground truth annotation in the 150ms neighborhood of the prediction, then we count a false positive(FP). Once every prediction has been compared with the manual labels, we count for each point of the ground truth annotation which has not been related to any prediction a false negative(FN).

Based on this, we calculate the following evaluation metrics:

- mean error $m$

- standard deviation of error $\sigma$

- sensitivity

$$Se = \frac{TP}{TP + FN} \qquad (5)$$

- positive predictive value

$$PPV = \frac{TP}{TP + FP} \qquad (6)$$

- F1-score

$$F1 = 2 \cdot \frac{Se \cdot PPV}{Se + PPV} \qquad (7)$$

*Se* indicates the algorithm's ability to identify true positives among all ground truth annotations, while *PPV* quantifies the algorithm's precision in detecting annotations. Furthermore, the F1-score, defined as the harmonic mean of *Se* and *PPV*, offers a unified assessment of the algorithm's performance. These metrics have been commonly used in the literature for the evaluation of ECG delineation algorithms [3, 4, 8, 40], and we use them to evaluate performance of our model.

## 4.2 Delineation performance on Arrhythmia

In this section we report how lack of arrhythmia diversity in training data affects evaluation results of our model using a test set with a diverse range of arrhythmia. We will describe detailed results in the table below, but first let us illustrate how the model's performance is directly affected by lack of diverse training data through various ECG signals from the PTB-XL dataset [48]. In Figs 6–8 we present delineation output of two models, namely model a), trained on LUDB data, and model b) trained using the internal dataset. In Fig 6, we observe that model a) fails to detect any P wave in a signal during sinus tachycardia. Looking at other

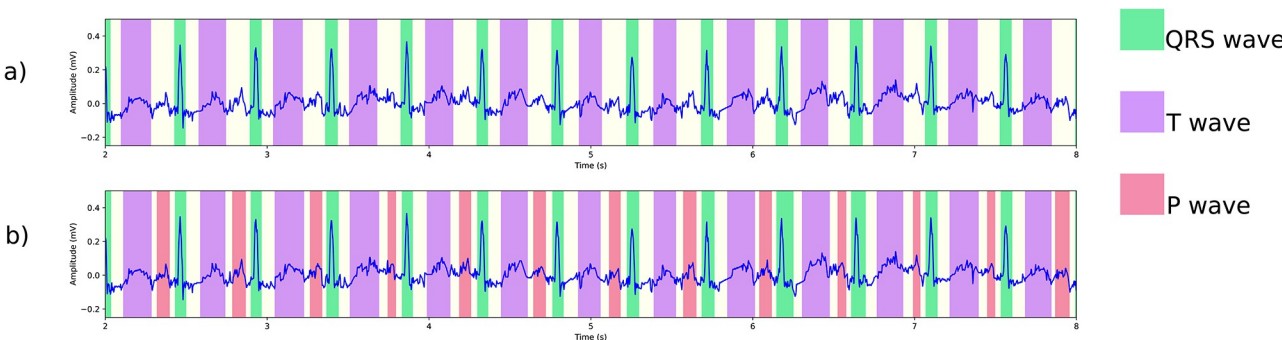

**Fig 6.** Delineation of an ECG showing sinus tachycardia (PTB-XL ECG-ID: 857) using two different models: (a) a model trained on LUDB, which is somewhat short on tachycardia samples, fails to detect the fairly obvious P waves; (b) a model trained on more diverse data with otherwise identical settings, performs much better.

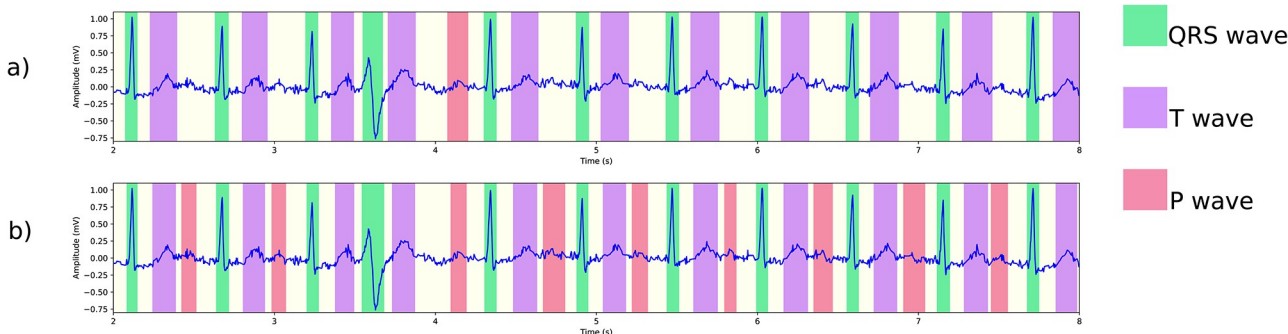

**Fig 7.** Delineation of an ECG showing sinus tachycardia and AVB1 (PTB-XL ECG-ID: 3337) using two different models: (a) a model trained on LUDB delineates all QRS complexes and T waves, including the premature ventricular complex, correctly, but misses P waves that are in shorter RR intervals; (b) a model trained on more diverse data with otherwise identical settings, finds all P waves.

arrhythmias, specifically AVB1 in Fig 7, both models perform well in detection of QRS complexes and T waves, but model a) has trouble with identifying the P waves. As a final example, Fig 8 presents the performance of the two models on a signal with bundle branch block and premature ventricular complexes. In this case, both models detect all waves correctly according to the standard of AAMI. However, model a) underestimates the width of the QRS complex: it puts the S wave offset well before the J point. We note that this type of defect is only visible in the mean error. The other metrics do not reveal this type of flaw. We will now check these phenomena systematically by delineating the test set of the internal dataset, which contains a wide range of arrhythmia. We point out that model a) and b) both perform well on QTDB (and LUDB); performance on these benchmark sets is addressed in the next section.

To assess the model's ability to handle signals with diverse arrhythmias, we measure the F1-scores separately for each of the following arrhythmia types: normal sinus rhythm (NSR), sinus tachycardia (ST), bundle branch block (BBB), first degree atrioventricular block (AVB1), atrial fibrillation (AFIB), atrial flutter (AFL) and ventricular tachycardia (VT). We also examine how the arrhythmia distribution of the training set can affect the delineation performance. For this, we train a separate segmentation model using LUDB as the only training set and compare the resulting delineation performance. LUDB has often been used in previous studies [7, 29] for training a segmentation model for the purposes of delineation. Here, we follow the same approach but test it on the internal dataset in order to measure performance for different

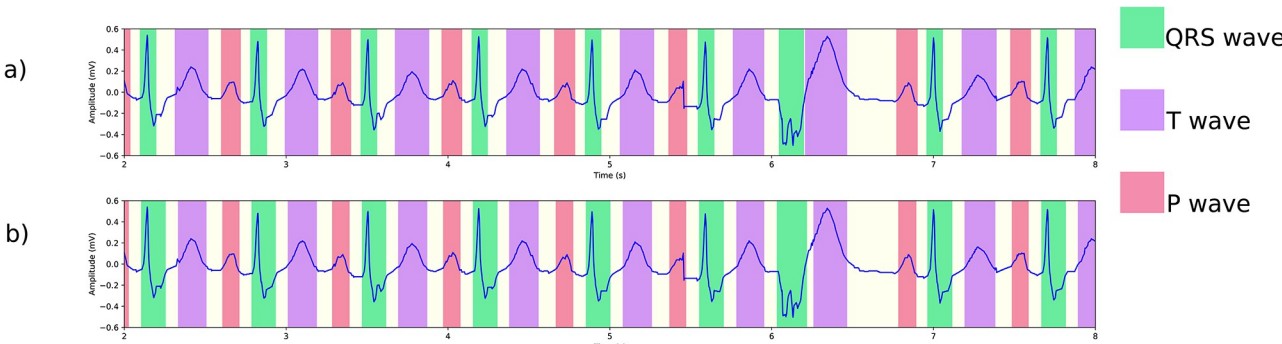

**Fig 8.** Delineation of an ECG showing bundle branch block and premature ventricular complex (PTB-XL ECG-ID: 287) using two different models: (a) a model trained on LUDB detects all waves correctly, but underestimates the width of all QRS complexes with the exception of the PVC; (b) a model trained on more diverse data with otherwise identical settings, detects the onsets and offsets accurately.

**Table 3. Arrhythmia dependence of onset and offset delineation performance on a test set comprised of diverse arrhythmia.** The training data strongly affects the models' performance as highlighted in the bold-faced F1-scores: scores can drop more than 15%.

| Training | Rhythm | F1-scores (%) | | | | | |
|---|---|---|---|---|---|---|---|
| | | **P onset** | **P offset** | **QRS onset** | **QRS offset** | **T onset** | **T offset** |
| Trained on LUDB (limited diversity) | NSR | 99.84 | 99.84 | 99.83 | 99.84 | 99.97 | 99.97 |
| | ST | **81.54** | **81.54** | 99.93 | 99.93 | 97.59 | 98.83 |
| | BBB | 98.89 | 98.89 | 99.94 | 99.94 | 99.89 | 99.94 |
| | AVB1 | **90.53** | **90.97** | 99.82 | 99.82 | 100.00 | 100.00 |
| | AFIB | - | - | 99.29 | 99.29 | 97.92 | 97.60 |
| | AFL | - | - | 99.21 | 99.21 | 92.67 | 93.00 |
| | VT | - | - | 91.04 | 91.11 | **78.78** | **78.63** |
| | All | 90.37 | 90.46 | 99.45 | 99.46 | 97.42 | 97.54 |
| Trained on diverse dataset | NSR | 99.69 | 99.69 | 99.78 | 99.81 | 99.95 | 99.95 |
| | ST | **97.19** | **97.19** | 99.91 | 99.91 | 99.90 | 99.94 |
| | BBB | 99.00 | 99.00 | 99.94 | 99.94 | 99.88 | 99.89 |
| | AVB1 | **95.93** | **95.93** | 99.84 | 99.84 | 100.00 | 100.00 |
| | AFIB | - | - | 99.54 | 99.54 | 99.56 | 99.54 |
| | AFL | - | - | 98.97 | 98.97 | 98.56 | 97.57 |
| | VT | - | - | 97.83 | 96.84 | **94.49** | **94.61** |
| | All | 96.47 | 96.46 | 99.71 | 99.69 | 99.67 | 99.63 |

arrhythmias. For a reliable comparison, each evaluation is repeated 20 times and the average score is reported.

Table 3 shows the F1-scores for the onset and offset delineation. From the results, we see that the model trained on the internal dataset can accurately delineate signals of each of the identified arrhythmia types. The F1-scores are mostly above 0.99, and all above 0.97 except for VT and P waves for AVB1. By contrast, the model trained on LUDB shows a much higher variation across different arrhythmia types. For normal sinus rhythm, exceptional F1-scores (over 0.99) are achieved. However, the effect of arrhythmia in delineation accuracy is noticeable in the F1-scores for P waves during ST and AVB1, and T waves during ST, AFIB, AFL, and VT.

## 4.3 External validation on QTDB and LUDB

In this section we will see that the improved performance in case of arrhythmia does not come at the cost of a good benchmark score. Our algorithm's ability to handle previously unseen signals is verified using the public datasets QTDB and LUDB. For LUDB, we compared our results with delineation algorithms using wavelets [4] and previous deep segmentation based methods [8, 29]. The evaluation was conducted on LUDB signals from leads I and II. Regarding QTDB, we benchmark against wavelet-based techniques [4, 40] and a recent deep learning approach [7]. Detailed results and comparisons with existing methods are shown in Table 4. Note that there are some discrepancies in annotation format which we shall elaborate in the following section.

**4.3.1 Discrepancies in annotation format.** The study utilizes three databases: internal, LUDB, and QTDB, each annotated by different cardiologist experts. These databases differ not only in the number of leads but also in the duration of recordings. Specifically, the internal and LUDB databases contain short 10-second signals with complete waveform boundary annotations except possibly for one or two initial and final cardiac cycles [8]. In contrast, QTDB consists of 15-minute recordings with annotations included for selected beats. Notably, the annotation format of QTDB, as discussed in [3, 40], does not allow us to measure the exact

**Table 4. Comparison of delineation performance on QTDB and LUDB.** For a direct comparison, we have considered the results of Moskalenko et al. [8] which uses single lead input, namely lead II. N/A: not applicable, N/R: not reported. This table shows that the performance of model, trained on diverse arrhythmia, has a performance that is comparable to that of other recent models.

| Database | Method | Metrics | P onset | P offset | QRS onset | QRS offset | T onset | T offset |
|---|---|---|---|---|---|---|---|---|
| QTDB | Di Marco et al. [40] | Se (%) | 98.15 | 98.15 | 100.0 | 100.0 | - | 99.77 |
| | | PPV (%) | 91.00 | 91.00 | N/A | N/A | | 97.76 |
| | | m ± σ (ms) | -4.5 ± 13.4 | -2.5 ± 13.0 | -5.1 ± 7.2 | 0.9 ± 8.7 | | 1.3 ± 18.6 |
| | Kalyakulina et al. [4] | Se (%) | 97.46 | 97.53 | 98.42 | 98.42 | - | 96.16 |
| | | PPV (%) | 97.86 | 97.93 | 98.24 | 98.24 | | 94.87 |
| | | m ± σ (ms) | 3.5 ± 13.8 | 3.4 ± 12.7 | -5.1 ± 6.6 | 4.7 ± 9.5 | | 13.4 ± 18.5 |
| | Chen et al. [7] | Se (%) | 99.58 | 99.78 | 100.0 | 100.0 | - | 98.63 |
| | | PPV (%) | N/R | N/R | N/A | N/A | | N/R |
| | | m ± σ (ms) | -0.6 ± 20.9 | 4.9 ± 19.5 | 1.3 ± 11.4 | 3.8 ± 18.8 | | 7.4 ± 32.5 |
| | Our Method | Se (%) | 96.51 | 96.55 | 100.0 | 100.0 | - | 97.50 |
| | | PPV (%) | 97.94 | 97.97 | N/A | N/A | | 95.31 |
| | | m ± σ (ms) | 13.0 ± 16.1 | -3.3 ± 18.5 | 4.1 ± 11.2 | 2.8 ± 17.3 | | -0.4 ± 35.1 |
| LUDB | Kalyakulina et al. [4] | Se (%) | 98.46 | 98.46 | 99.61 | 99.61 | - | 98.03 |
| | | PPV (%) | 96.41 | 96.41 | 99.87 | 99.87 | | 98.84 |
| | | m ± σ (ms) | -2.7 ± 10.2 | 0.4 ± 11.4 | -8.1 ± 7.7 | 3.8 ± 8.8 | | 5.7 ± 15.5 |
| | Sereda et al. [29] | Se (%) | 95.20 | 95.39 | 99.51 | 99.50 | 97.95 | 97.56 |
| | | PPV (%) | 82.66 | 82.59 | 98.17 | 97.96 | 94.81 | 94.96 |
| | | m ± σ (ms) | 2.7 ± 21.9 | -7.4 ± 28.6 | 2.6 ± 12.4 | -1.7 ± 14.1 | 8.4 ± 28.2 | -3.1 ± 28.2 |
| | Moskalenko et al. [8] | Se (%) | 98.61 | 98.59 | 99.99 | 99.99 | 99.32 | 99.40 |
| | | PPV (%) | 95.61 | 95.59 | 99.99 | 99.99 | 99.02 | 99.10 |
| | | m ± σ (ms) | -4.1 ± 20.4 | 3.7 ± 19.6 | 1.8 ± 13.0 | -0.2 ± 11.4 | -3.6 ± 28.0 | -4.1 ± 35.3 |
| | Our Method | Se (%) | 98.16 | 98.20 | 99.67 | 99.97 | 99.82 | 99.63 |
| | | PPV (%) | 96.39 | 96.36 | 99.29 | 99.59 | 99.66 | 99.42 |
| | | m ± σ (ms) | 7.4 ± 14.1 | -1.8 ± 9.9 | 6.1 ± 10.5 | 2.0 ± 10.7 | 3.0 ± 25.2 | 4.5 ± 24.4 |

PPV value. In fact, when there is no annotation, we cannot decide whether the waveform is not present or the annotation is simply not included. To address this, we adopt the approach from [3, 40] and treat an absent manual annotation on a predicted beat as a non-included annotation. To ensure consistency with [4, 40], we select the lead with the lowest error for each boundary point.

## 4.4 Reduced false P wave predictions

Arrhythmia classification guidance was presented in the Arrhythmia Classification Guidance Section 3.8 as a method to reduce the number of false P wave detections which occur frequently during atrial fibrillation and flutter events. To evaluate its effectiveness, we compared the number of false positive P wave predictions generated by models trained with and without classification guidance. Table 5 shows the results, including the PPV and Se scores for the entire test set as a reference.

## 4.5 Examples of delineation results

This section presents examples of our algorithm's delineation outcomes on the MIT-BIH arrhythmia database [38]. We have chosen multiple instances of arrhythmia to showcase how our algorithm handles them, as depicted in Fig 9. Other challenges are shown in Fig 10, including noise, baseline wander, and loss of signal.

**Table 5. Number of false positive P annotations for AFIB and AFL.** The *PPV* and *Se* scores for the entire test set are shown for reference. The values are averaged over 20 runs.

| | AFIB (1437 beats) | | AFL (540 beats) | | All (14418 beats) | | | |
|---|---|---|---|---|---|---|---|---|
| | False Positives | | False Positives | | *PPV* (Precision) | | *Se* (Recall) | |
| | P onset | P offset | P onset | P offset | P onset | P offset | P onset | P offset |
| Trained w/o classification | 62.35 | 62.35 | 34.25 | 34.25 | 97.53 | 97.52 | **95.43** | **95.43** |
| Trained w/ classification | **13.85** | **13.85** | **1.85** | **1.9** | **98.70** | **98.69** | 95.31 | 95.31 |

Our method provides accurate delineation in all the presented examples, highlighting its versatility in several aspects. First, with the exception of signal resampling, no additional signal processing techniques were used to achieve the results. Second, due to the convolutional nature of the segmentation model, the algorithm can accommodate signals of varying lengths. This greatly enhances its utility, particularly in the context of Holter recordings containing potential arrhythmias, allowing for the algorithm's application to windows of sizes chosen for convenience. Our pytorch implementation segments and delineates an ECG record of 30 minutes in under 2s-3s on an Ubuntu machine with 64GB DRAM equipped with an NVIDIA 3080Ti with 12GB memory. The model itself uses a little under $20 \cdot 10^6$ parameters, and needs about 80 MB of memory. In particular, this is both suitable for real time analysis and the intended application of the analysis of long Holter recordings. Finally, it is worth noting that no parameter tuning was necessary for the delineation when applied to the MIT-BIH arrhythmia database.

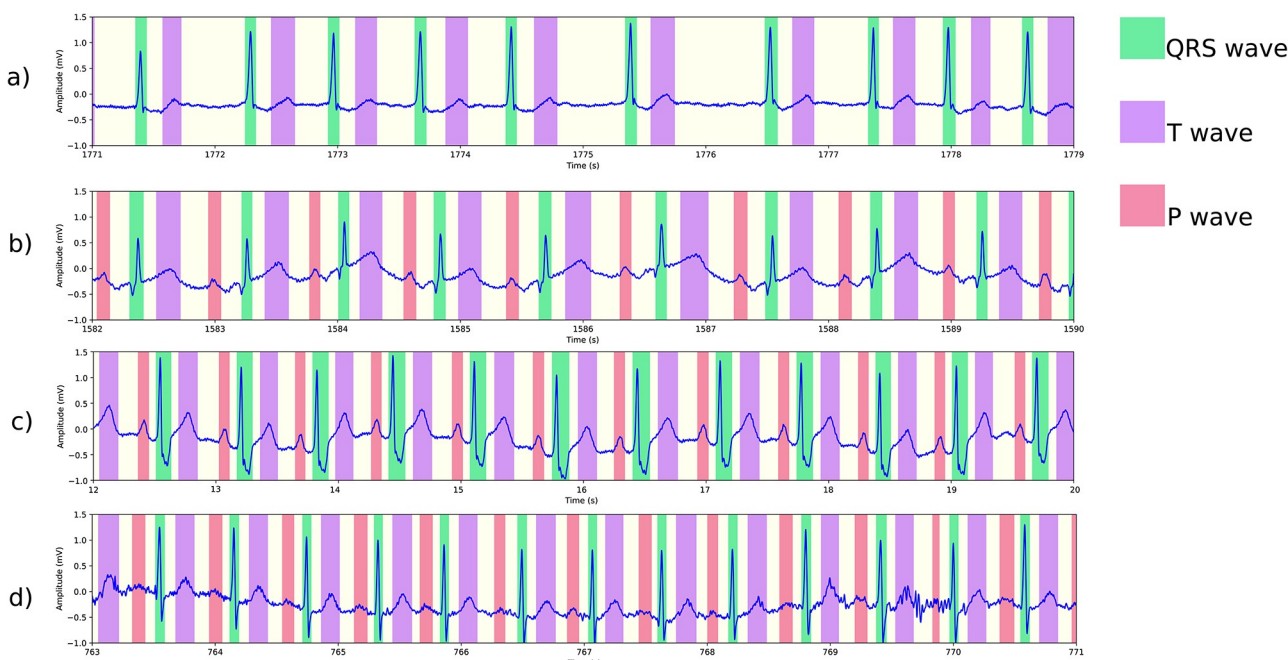

**Fig 9. Segmentation results on the MIT-BIH arrhythmia database.** (a) Atrial fibrillation in record 221. The small bumps are not misidentified as P waves, and we have observed the same correct behavior in the presence of atrial flutter. (b) First degree atrioventricular block in record 228, with correct detection of longer-than-normal PR intervals. (c) Bundle branch block in record 212, featuring a wide QRS complex. (d) Sinus tachycardia in record 209, with heart rate slightly over 100 bpm.

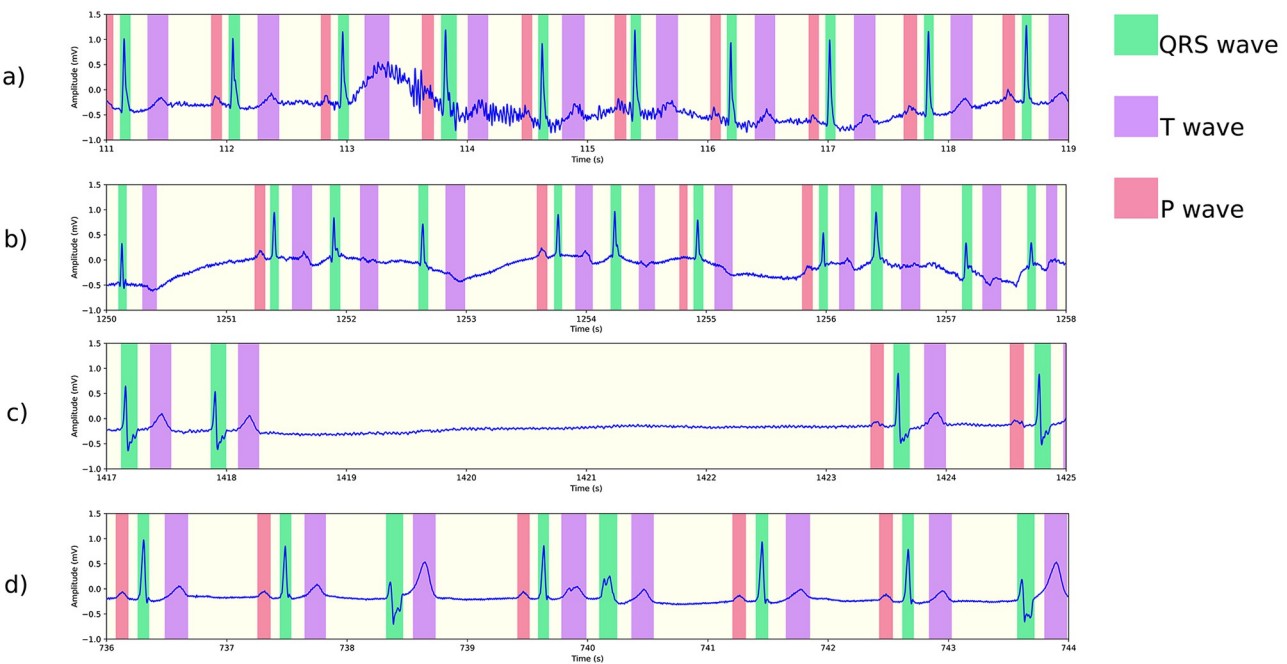

**Fig 10. More segmentation results on the MIT-BIH arrhythmia database.** (a) Normal sinus rhythm in record 101, with baseline oscillations and noise. (b) The onset of an episode of atrial flutter in record 222. The early signal displays normal sinus rhythm with PAC, and P waves being detected. Later, atrial flutter without P waves is observed. (c) An episode of loss of signal in record 232. (d) Ventricular trigeminy in record 201.

## 5 Discussion

### 5.1 Delineation of ECGs with arrhythmia

In Table 3 we see that there is a significant difference between the LUDB trained model and the model trained on internal data with regard to ECGs with certain arrhythmia. There is almost no performance difference for NSR and BBB, but for the arrhythmias that are not so well-represented in LUDB, the difference is striking. For example, in LUDB, 15 recordings represent signals with atrial fibrillation, while only three recordings with atrial flutter and four recordings with sinus tachycardia are available [10]. The performance drops especially in the latter case. Fig 3 shows that in some cases of tachycardia all P waves can be missed by an improperly trained model. Similar problems can occur in AVB1. This brings us to another problem; the LUDB trained model has a high number of false positive P waves for AFIB and AFL. Without testing the model on a dataset that has a balanced distribution of arrhythmias, it is difficult to identify such failure cases. Overall, the results from Table 3 highlight the importance of using a well-curated dataset that encompasses a broad range of arrhythmias commonly seen in clinical practice for developing and validating an ECG delineation algorithm. For completeness, we reiterate that model a), trained on LUDB, although it has poor performance on tachycardia, still performs well on the benchmark QTDB (and of course on LUDB). The model trained on more diverse data has much better performance in cases of arrhythmia, while retaining a good performance on the standard benchmark tests as we will discuss next.

### 5.2 Performance on standard benchmarks

From Table 4 we see that our method shows performance comparable to existing methods in terms of accuracy and error metrics. Particularly on QTDB, our method shows high

performance in delineating P wave onsets and offsets, achieving a *PPV* of over 97.9%, outperforming the methods we compared against. In the case of LUDB, our method's strength lies in accurate T wave delineation, with both *Se* and *PPV* exceeding 99.4%, an improvement over other methods. Taken together, these results underscore the consistent accuracy of our proposed delineation algorithm across various waveforms. Our method's weakest point is observed in the standard deviation of error ($\sigma$), particularly noticeable for the T offset of QTDB signals. In fact, we can observe from Table 4 that deep learning-based methods tend to exhibit higher $\sigma$ compared to wavelet-based methods. This also aligns with the observations of Jimenez-Perez et al. [6], where their deep learning-based delineation also reported a $\sigma$ larger than 30ms for T offset delineation in QTDB. We also observe that the onset errors for P and QRS are shifted positively while the standard deviation remains relatively similar to other methods, which may partially be an artifact of the independent annotations for training and test data.

It is worth noting that the comparable performance on the public datasets has been achieved by training exclusively on the internal dataset. This is important as it implies the high generalization ability of the proposed algorithm and deep learning based methods in general. As noted in [6, 8], the ability to handle unseen signals without the need for additional tuning of parameters is a key advantage of deploying a deep learning model compared to waveletbased methods. By using a private dataset as opposed to a portion of either QTDB or LUDB for training, we have made a clear demonstration of the effectiveness at which deep segmentation models can be applied to diverse scenarios.

### 5.3 Arrhythmia guidance

The results on arrhythmia classification guidance, Table 5, indicate a significant reduction in false positives for both atrial fibrillation and atrial flutter. When compared to the total number of beats corresponding to the same rhythm type (indicated in the header row of Table 5), the number of false positives for the classification guided model is less than 1%. The reduction in false predictions is reflected in the improved *PPV* scores for P waves belonging to the entire test set, while recall scores only decreased slightly. As a result, the total F1-score increased from 96.47% to 96.97%. From the results, we conclude that the classification guided strategy can be effective in reducing false P wave predictions during AFIB and AFL episodes while maintaining the overall delineation quality.

We point out that there are two mechanisms via which classification guidance can affect delineation performance. Firstly, the weights of the segmentation model are influenced by training with the hybrid loss function (4) which combines both segmentation and classification losses. Secondly, the P wave segmentation output is suppressed according to the output of the classification branch. To illustrate this, delineation results for AFIB in Fig 11 show outcomes using a model trained with arrhythmia classification guidance: (a) without P wave suppression and (b) with P wave suppression. The positive output from the classification branch triggers P wave suppression, removing multiple false P wave predictions from the segmentation output. Note that this direct suppression of P wave segmentation is primarily applicable to short signals without rhythm changes. In ECGs with rhythm changes, rhythms cannot be unambiguously classified. A potential avenue for improvement could involve combining with accurate per-beat classification, which would require more refined training data.

### 5.4 Limitations and future directions

The current delineation algorithm has limitations which should be considered when applying it to clinical practice. Firstly, it can only detect a single P wave within an RR interval due to the

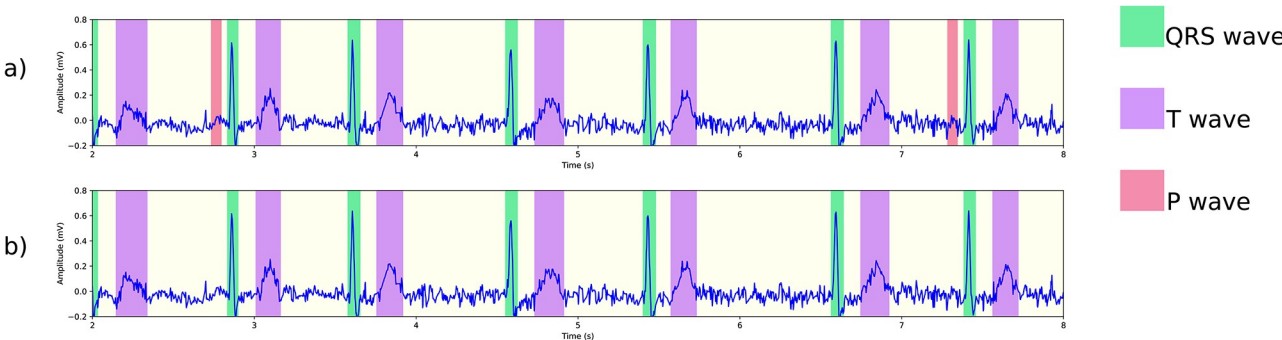

**Fig 11. Delineation of atrial fibrillation sample (PTB-XL ECG-ID: 5634) using a model trained with arrhythmia classification guidance (a) without P wave suppression and (b) with P wave suppression.**

post-processing step (Post-processing Section 3.7), where one P wave segment is selected per RR interval. This design is particularly effective for mitigating noise, but limits applicability to abnormal rhythms, such as second or third-degree atrioventricular blocks, where multiple P waves may precede a QRS complex. Secondly, the assumption of non-overlapping waveforms (P, QRS, and T) in the algorithm's output is a further restriction; overlapping waveforms can for example occur in first, second or third-degree atrioventricular blocks. We note that these limitations are not unique to our algorithm but are common in deep learning based delineation approaches [7, 8].

To address these limitations, future work may incorporate flexibility into a delineation algorithm's output by allowing for the detection of multiple P waves within RR intervals, implementing multi-label classification techniques or separate models for QRS/T waves (depolarization/repolarization of ventricles) and P waves (depolarization of atria). Moreover, advanced data augmentation techniques should be investigated to accommodate other arrhythmias for which collecting annotated data may be challenging or impractical. These future directions aim to enhance delineation performance and widen its scope of application in clinical settings.

## 6 Conclusion

One of the main challenges in ECG delineation is to accurately identify and delineate waveforms within irregular cardiac rhythms. This study aimed to develop a deep learning-based segmentation model capable of detecting the onsets and offsets of P, QRS, and T waves in signals with potential arrhythmias. By evaluating on the internal dataset, we have highlighted the impact of arrhythmias on delineation quality. We observed significant drops in F1-scores for waveform boundary detection, particularly with arrhythmias such as ST, AVB1, AFIB, AFL, and VT, with reductions of up to 15% in certain cases, emphasizing the need to account for arrhythmias when developing and evaluating segmentation models for ECG analysis. To address this, we experimented with training on a diverse dataset and employing a post-processing strategy that can handle noise during the final delineation step. Furthermore, we assessed generalization capability through experiments on the QTDB and LUDB datasets. Our model demonstrated strong performance on the LUDB dataset, achieving Se and PPV scores above 99% for QRS and T wave boundaries, and above 98% and 96% respectively for P waves, showing comparable performance without direct training on LUDB signals. Overall, our study shows a deep learning based segmentation model to be a versatile tool for delineation which can be highly adaptive to various situations, while addressing the challenge of accurately

delineating waveforms in abnormal cardiac rhythms. Future research and development could focus on broadening the scope of automatic delineation to encompass a wider range of arrhythmias, through more manual annotations or advanced data augmentation techniques.

## Supporting information

**S1 Appendix.**
(PDF)

## Author Contributions

**Conceptualization:** Chankyu Joung, Mijin Kim.

**Data curation:** Mijin Kim, Seong-Ho Kong, Seung-Young Oh, Won Kyeong Jeon, Myung-Jin Cha.

**Formal analysis:** Chankyu Joung.

**Funding acquisition:** Joong-Sik Hong, Woong Kook, Myung-Jin Cha.

**Investigation:** Chankyu Joung, Mijin Kim.

**Methodology:** Chankyu Joung, Mijin Kim, Myung-Jin Cha.

**Project administration:** Seung-Young Oh, Jae-hu Jeon.

**Resources:** Taejin Paik.

**Software:** Chankyu Joung, Taejin Paik.

**Supervision:** Seong-Ho Kong, Jae-hu Jeon, Joong-Sik Hong, Wan-Joong Kim, Woong Kook.

**Validation:** Chankyu Joung, Taejin Paik.

**Visualization:** Otto van Koert.

**Writing – original draft:** Chankyu Joung, Mijin Kim, Otto van Koert.

**Writing – review & editing:** Chankyu Joung, Taejin Paik, Woong Kook, Otto van Koert.

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
