## [Decision Letter · Decision Letter 0]

27 Feb 2024

PONE-D-23-38359Deep Learning based ECG Segmentation for Delineation of Diverse ArrhythmiasPLOS ONE

Dear Dr. van Koert,

Thank you for submitting your manuscript to PLOS ONE. After careful consideration, we feel that it has merit but does not fully meet PLOS ONE’s publication criteria as it currently stands. Therefore, we invite you to submit a revised version of the manuscript that addresses the points raised during the review process.

 perceived impact.

We look forward to receiving your revised manuscript.

Kind regards,

Mohamed Hammad, Ph.D.

Academic Editor

PLOS ONE

Journal Requirements:

Reviewers' comments:

Reviewer's Responses to Questions

**Comments to the Author**

1. Is the manuscript technically sound, and do the data support the conclusions?

Reviewer #1: Yes

Reviewer #2: Partly

2. Has the statistical analysis been performed appropriately and rigorously? 

Reviewer #1: Yes

Reviewer #2: No

3. Have the authors made all data underlying the findings in their manuscript fully available?

Reviewer #1: Yes

Reviewer #2: Yes

4. Is the manuscript presented in an intelligible fashion and written in standard English?

Reviewer #1: Yes

Reviewer #2: Yes

5. Review Comments to the Author

Reviewer #1: 1- The abstract did not explain the working database, nor the standards adopted to measure the model’s efficiency, nor the steps of the algorithm in a simplified manner

2- The size of the tables is very small

3-Equations are not numbered

4-Research included in related works is not arranged according to years of publication (from oldest to newest)

5- Explaining the researcher’s contributions in clear steps. In addition to placing a paragraph at the end of the introduction section that explains the structure of the research in general

6- Prepare a diagram showing the structure of the proposed model

7-Conclusions: No numerical values were indicated for the results to be discussed concretely, in addition to developing a future plan for those working in this field

8-Adding recent research as references published in the year 2023, in addition to unifying the format of references

Reviewer #2: this paper presents an important investigation into the challenges of ECG delineation in the presence of diverse arrhythmias and proposes strategies to improve segmentation model performance.

- The abstract provides a clear overview of the paper's objectives and contributions. However, it would benefit from briefly mentioning the methodology and key findings.

- Introduction lacks a clear statement of the paper's objectives and contributions.

- While the authors have provided extensive information about the dataset, additional details regarding the specific criteria used for arrhythmia classification and the annotation process for waveform boundaries would enhance the clarity and rigor of the methodology.

-Provide detailed information on how arrhythmia types were identified and classified within the internal dataset.

- Elaborate on the methodology employed for annotating the onsets and offsets of P, QRS, and T waves in the internal dataset. Describe any quality control measures implemented to ensure accuracy and consistency in the annotations.

- Provide additional details on the preprocessing steps applied to the training data, particularly regarding the augmentation techniques used to simulate physiological noise.

- Include a brief discussion on the limitations of the proposed methodology, such as potential biases introduced by the dataset composition or model architecture choices. Additionally, suggest potential avenues for future research to address these limitations and further enhance the algorithm's performance.

- The authors have appropriately chosen evaluation metrics commonly used in the literature for ECG delineation algorithms. However, it would be beneficial to provide a brief explanation or definition of each metric

- The authors have effectively demonstrated the impact of arrhythmia diversity on delineation performance using both the internal dataset and LUDB. The systematic comparison of F1-scores for various arrhythmia types provides valuable insights. However, including specific examples or case studies illustrating delineation outcomes for different arrhythmias could further enrich the discussion.

- it would be helpful to provide a brief discussion on any observed discrepancies or challenges encountered during the validation process, particularly regarding differences in annotation formats and their impact on performance evaluation.

- The evaluation of arrhythmia classification guidance to reduce false positive P wave detections is a valuable contribution. However, the discussion could be enhanced by providing insights into the underlying reasons for the observed improvements and discussing any potential limitations or trade-offs associated with this approach.

- Related work Section provides a concise summary of rule-based methods and classical machine learning approaches for ECG delineation. However, it would be beneficial to briefly mention the limitations or challenges associated with these traditional techniques, such as their dependence on handcrafted features and susceptibility to noise.

- it would be helpful to include a brief explanation of the rationale behind using CNN architectures for segmentation tasks, emphasizing their ability to automatically learn hierarchical features from raw data.

6. PLOS authors have the option to publish the peer review history of their article (what does this mean?). If published, this will include your full peer review and any attached files.

Reviewer #1: No

Reviewer #2: No

---

## [Author Response · Author response to Decision Letter 0]

10 Apr 2024

First of all we would like to thank the reviewers for their careful reading and for their suggestions. We have revised the paper according to their suggestions. We will now briefly address each comment:

Points raised by reviewer #1:

1- The abstract did not explain the working database, nor the standards adopted to measure the model’s efficiency, nor the steps of the algorithm in a simplified manner:

We have expanded the abstract to address these points.

2- The size of the tables is very small

We have increased the size of all tables.

3-Equations are not numbered

Fixed

4-Research included in related works is not arranged according to years of publication (from oldest to newest)

We believe that, after unifying and cleaning up the references (as in suggestion 8 of the reviewer), the ordering is now correct. The order of appearance is first by topic and then by year.

5- Explaining the researcher’s contributions in clear steps. In addition to placing a paragraph at the end of the introduction section that explains the structure of the research in general

We have included a brief summary of the key contributions at the end of the introduction.

6- Prepare a diagram showing the structure of the proposed model

A flow diagram showing the structure of the model has been included in Fig 2 appearing at the end of the introduction.

7-Conclusions: No numerical values were indicated for the results to be discussed concretely, in addition to developing a future plan for those working in this field

We have rewritten part of the conclusion to address these points. Regarding the development of a future plan, we have written most in a new section “Limitations and Future Directions” appearing just before the conclusion (section 5.4)

8-Adding recent research as references published in the year 2023, in addition to unifying the format of references

We have included additional references, including some from 2023, and cleaned up the references.

Points raised by reviewer #2:

- The abstract provides a clear overview of the paper's objectives and contributions. However, it would benefit from briefly mentioning the methodology and key findings.

The extended abstract now mentions these points briefly.

- Introduction lacks a clear statement of the paper's objectives and contributions.

The end of the introduction now contains a list of key contributions, following the objectives.

- While the authors have provided extensive information about the dataset, additional details regarding the specific criteria used for arrhythmia classification and the annotation process for waveform boundaries would enhance the clarity and rigor of the methodology.

We have added a paragraph (lines 196-207) and an appendix to address these points.

-Provide detailed information on how arrhythmia types were identified and classified within the internal dataset.

This is now also explained in this additional paragraph.

- Elaborate on the methodology employed for annotating the onsets and offsets of P, QRS, and T waves in the internal dataset. Describe any quality control measures implemented to ensure accuracy and consistency in the annotations.

The methology is described in the additional paragraph. The appendix presents statistics on the consistency of lead I and lead II annotation. These statistics give an indication of the accuracy and consistency of the annotation depending on the type of wave.

- Provide additional details on the preprocessing steps applied to the training data, particularly regarding the augmentation techniques used to simulate physiological noise.

Formulas for the augmentation methods are now provided in Formulas (1) and (2). The surrounding text has been expanded to provide additional detail.

- Include a brief discussion on the limitations of the proposed methodology, such as potential biases introduced by the dataset composition or model architecture choices. Additionally, suggest potential avenues for future research to address these limitations and further enhance the algorithm's performance.

This has been addressed in the additional section 5.4 on Limitations and Future Directions (lines 538-556)

- The authors have appropriately chosen evaluation metrics commonly used in the literature for ECG delineation algorithms. However, it would be beneficial to provide a brief explanation or definition of each metric

We have added lines 373-376 to explain these metrics.

- The authors have effectively demonstrated the impact of arrhythmia diversity on delineation performance using both the internal dataset and LUDB. The systematic comparison of F1-scores for various arrhythmia types provides valuable insights. However, including specific examples or case studies illustrating delineation outcomes for different arrhythmias could further enrich the discussion.

We have included three examples in Figures 6, 7 and 8 by annotating samples from the PTB-XL database. These samples were chosen based on the increased error rate in certain arrhythmias and are indicative of the biases Deep Learning models can obtain.

- it would be helpful to provide a brief discussion on any observed discrepancies or challenges encountered during the validation process, particularly regarding differences in annotation formats and their impact on performance evaluation.

This has been done in the additional section 4.3.1, lines 429-435.

- The evaluation of arrhythmia classification guidance to reduce false positive P wave detections is a valuable contribution. However, the discussion could be enhanced by providing insights into the underlying reasons for the observed improvements and discussing any potential limitations or trade-offs associated with this approach.

An explicit example of arrhythmia guidance suppression is now provided in Fig 11. An additional paragraph, lines 525-537, contains a discussion of potential limitations and drawbacks.

- Related work Section provides a concise summary of rule-based methods and classical machine learning approaches for ECG delineation. However, it would be beneficial to briefly mention the limitations or challenges associated with these traditional techniques, such as their dependence on handcrafted features and susceptibility to noise.

We thank the reviewer for pointing this out and have added these observations in lines 117-121.

- it would be helpful to include a brief explanation of the rationale behind using CNN architectures for segmentation tasks, emphasizing their ability to automatically learn hierarchical features from raw data.

We have included these observations in lines 127-130

We thank the reviewers once again.

Sincerely yours,

Otto van Koert

---

## [Decision Letter · Decision Letter 1]

22 Apr 2024

Deep Learning based ECG Segmentation for Delineation of Diverse Arrhythmias

PONE-D-23-38359R1

Dear Dr. van Koert,

We’re pleased to inform you that your manuscript has been judged scientifically suitable for publication and will be formally accepted for publication once it meets all outstanding technical requirements.

Kind regards,

Mohamed Hammad, Ph.D.

Academic Editor

PLOS ONE

Additional Editor Comments (optional):

Reviewers' comments:

Reviewer's Responses to Questions

**Comments to the Author**

1. If the authors have adequately addressed your comments raised in a previous round of review and you feel that this manuscript is now acceptable for publication, you may indicate that here to bypass the “Comments to the Author” section, enter your conflict of interest statement in the “Confidential to Editor” section, and submit your "Accept" recommendation.

Reviewer #1: All comments have been addressed

Reviewer #2: All comments have been addressed

2. Is the manuscript technically sound, and do the data support the conclusions?

Reviewer #1: Yes

Reviewer #2: Yes

3. Has the statistical analysis been performed appropriately and rigorously? 

Reviewer #1: Yes

Reviewer #2: Yes

4. Have the authors made all data underlying the findings in their manuscript fully available?

Reviewer #1: Yes

Reviewer #2: Yes

5. Is the manuscript presented in an intelligible fashion and written in standard English?

Reviewer #1: Yes

Reviewer #2: Yes

6. Review Comments to the Author

Reviewer #1: (No Response)

Reviewer #2: Dear Authors,

Congratulations!

All comments have been addressed.

7. PLOS authors have the option to publish the peer review history of their article (what does this mean?). If published, this will include your full peer review and any attached files.

Reviewer #1: No

Reviewer #2: No

---

## [Editor Report · Acceptance letter]

10 May 2024

PONE-D-23-38359R1 

PLOS ONE

Dear Dr. van Koert, 

I'm pleased to inform you that your manuscript has been deemed suitable for publication in PLOS ONE. Congratulations! Your manuscript is now being handed over to our production team.

Kind regards, 

on behalf of

Dr. Mohamed Hammad 

Academic Editor

PLOS ONE